# Arteriovenous Fistula Flow Dysfunction Surveillance: Early Detection Using Pulse Radar Sensor and Machine Learning Classification

**DOI:** 10.3390/bios11090297

**Published:** 2021-08-26

**Authors:** Cheng-Hsu Chen, Teh-Ho Tao, Yi-Hua Chou, Ya-Wen Chuang, Tai-Been Chen

**Affiliations:** 1Division of Nephrology, Department of Internal Medicine, Taichung Veterans General Hospital, Taichung City 40705, Taiwan; cschen@vghtc.gov.tw (C.-H.C.); colaladr@yahoo.com.tw (Y.-W.C.); 2Department of Life Sciences, Tunghai University, Taichung City 40724, Taiwan; 3School of Medicine, China Medical University, Taichung City 40640, Taiwan; 4College of Medicine, National Chung Hsing University, Taichung City 40227, Taiwan; 5Finedar Biomedical Technology Co., Ltd., Hsinchu City 30069, Taiwan; 6Center of Hemodialysis, Department of Nursing, Taichung Veterans General Hospital, Taichung City 40705, Taiwan; cihua544@gmail.com; 7Department of Medical Imaging and Radiological Science, I-Shou University, Kaohsiung City 82445, Taiwan; 8Institute of Statistics, National Yang Ming Chiao Tung University, Hsinchu 30010, Taiwan

**Keywords:** arteriovenous fistula, SVM, harmonic ratio, vascular wall motion monitor

## Abstract

Vascular Access (VA) is often referred to as the “Achilles heel” for a Hemodialysis (HD)-dependent patient. Both the patent and sufficient VA provide adequacy for performing dialysis and reducing dialysis-related complications, while on the contrary, insufficient VA is the main reason for recurrent hospitalizations, high morbidity, and high mortality in HD patients. A non-invasive Vascular Wall Motion (VWM) monitoring system, made up of a pulse radar sensor and Support Vector Machine (SVM) classification algorithm, has been developed to detect access flow dysfunction in Arteriovenous Fistula (AVF). The harmonic ratios derived from the Fast Fourier Transform (FFT) spectrum-based signal processing technique were employed as the input features for the SVM classifier. The result of a pilot clinical trial showed that a more accurate prediction of AVF flow dysfunction could be achieved by the VWM monitor as compared with the Ultrasound Dilution (UD) flow monitor. Receiver Operating Characteristic (ROC) curve analysis showed that the SVM classification algorithm achieved a detection specificity of 100% at detection thresholds in the range from 500 to 750 mL/min and a maximum sensitivity of 95.2% at a detection threshold of 750 mL/min.

## 1. Introduction

The National Kidney Foundation Kidney Disease Outcomes Quality Initiative (NKF-K/DOQI) guidelines suggest two monitoring methods for VA flow surveillance, physical examination, and the measuring of arteriovenous VA flow through special equipment [1]. Guideline 13 of the NKF-K/DOQI specifically states that the underlying cause of AV access flow dysfunction is either stenosis or thrombosis. Hence, the purpose of flow dysfunction surveillance is to detect stenosis early on before the development of thrombosis, which then requires surgical intervention to replace the AV access.

Amongst the current monitoring instruments available, the Ultrasound Dilution (UD) flow monitoring instrument has been extensively studied. For example, the benefits of AV Fistula (AVF) flow surveillance using the UD measurement was studied [2], where the effectiveness of UD measurement in detecting stenosis and predicting thrombosis was investigated [3,4,5]. Our clinical experience regarding the application of the UD measurement technique shows that it is limited due to its cost, the disposable consumables used, and its dependence on the operator’s experience with the instruments. However, through the use of skillful operation techniques, it has been accepted as a reference standard for AV access flow surveillance.

Studies have been performed surrounding the use of optical sensor and machine learning algorithms for the detection of stenosis or flow dysfunction. However, either two finger Photoplethysmography (PPG) sensors were used, which may cause measurement uncertainties [6] or additional physiological measurements other than the PPG signal were required as input features for classification of the algorithm [7,8]. Phono-angiography employs a digital stethoscope to record bruit sounds in the AVF. However, four locations are required in order to search for the most probable stenosis site [9,10]. Therefore, for the most part, detection sensitivity and specificity of the aforementioned methods do not meet the requirements for them to be reliable monitoring instruments. A qualified monitoring technique should possess standardized diagnostic thresholds, as well as sufficient sensitivity and specificity for the detection of flow dysfunction.

The physical examination techniques for VA flow surveillance include inspection, palpation (pulse, thrill), and auscultation (bruit) [1]. One of the clinical indicators for VA flow dysfunction is alterations in the pulse characteristics such as weak and persistent pulses which are difficult to compress in the area of stenosis. Although recommended by the NKF-K/DOQI guidelines, the technique of pulse palpation relies upon the subjective experiences of the examiner. Studies performed on the hemodynamics of stenosis in arteries were reviewed [11]. In a stenotic blood vessel, turbulent blood flow is generated approximately 1.5 to 6.0 diameters downstream from the site of stenosis. Finite element numerical simulation was employed to investigate pulsatile and turbulent blood flow in an elastic artery with single as well as double stenosis, with the results showing that the displacement of the arterial walls in the pre-stenotic regions was higher than that in the post-stenotic regions [12]. Raminari et al. investigated the potential clinical application of ultrasound Tissue Doppler Imaging (TDI) of arterial wall motion in order to quantify simple wall motion indices in normal and diseased carotid arteries. Their results showed a wide variation in arterial wall motion indices across the stenotic region. However, their experimental data showed noticeable changes in the morphology of the diseased arterial wall motion waveforms when compared to those of normal arteries [13]. The aforementioned studies demonstrate that stenosis of the arterial blood vessel, which is the underlying cause of flow dysfunction, can be detected by analyzing the characteristic changes in the waveform of arterial wall motion.

In the present work, the spectral analysis technique was adopted by analyzing the variations in the spectrum of the distorted VWM waveform due to the AVF flow dysfunction. A machine learning algorithm based on Support Vector Machines (SVM) was chosen to classify the spectrum data of VWM monitoring data. Through a comparison with the corresponding UD flow measurements, the performance of the SVM classifier was evaluated in terms of its detecting sensitivity, specificity, and accuracy.

## 2. Materials and Methods

### 2.1. The Sensing Device and System

The VWM monitor consists of a pulse radar sensor, a Microcontroller Unit (MCU), as well as a data analysis and classification unit, as shown in Figure 1A. The pulse radar sensor serves the purpose of sensing the motion of the A-V fistula vessel wall [14,15]. The operation principle of the pulse radar sensor is described as follows. Damped sinusoidal pulses with a pulse duration of 4 ns and repetition frequency of 250 K Hz are generated in the pulse generator, with its input connected to the square wave generator. As shown in Figure 1B, a sequence of damped sinusoidal pulses is emitted by the transmit antenna towards the patient’s arm where the AVF is located. The damped sinusoidal pulses can be expressed by Equation (1), where A is the pulse envelope of $\tilde{X}\left(t\right)$ and $f_{p}$ is the carrier frequency of $\tilde{X}\left(t\right).$
(1)$$\tilde{X}\left(t\right)=Asin\left(2\pi f_{p}t\right)$$

The scattered pulses from the AVF wall to the receiving antenna is then expressed by Equation (2), where $R\left(t\right)\;$ is the distance between the antenna and the AVF wall, c (meter/sec) is the speed of light, and ∈ is the effective permittivity of the skin and subcutaneous tissue.
(2)$$\tilde{Y}\left(t\right)=\tilde{X}\left(t-\frac{2R\left(t\right)}{c/\sqrt{\in }}\right)$$

Due to variations in blood pressure, the AVF wall is displaced towards the radar (Figure 1C), with the distance expressed by Equation (3), where $R_{o}$ is the initial distance between the antenna and the AVF wall and $\Delta r\left(t\right)$ is the displacement of the AVF wall. Substitute Equation (3) into Equation (2) and the scattered pulses are expressed by Equation (4).
(3)$$R\left(t\right)=R_{o}-\Delta r\left(t\right)$$
(4)$$\tilde{Y}\left(t\right)=Asin\left(2\pi f_{p}\left(t-\frac{2\left(R_{o}-\Delta r\left(t\right)\right)}{\frac{c}{\sqrt{\in }}}\right)\right)$$

The reference pulse (shown in Figure 1A) with a time delay of $\tilde{X}\left(t\right)$ can be expressed.
(5)$$\tilde{X}\left(t-\tau \right)=Asin\left(2\pi f_{p}\left(t-\tau \right)\right)$$

The received signal $\tilde{Y}$ and the delayed reference pulse $\tilde{X}\left(t-\tau \right)$ is then mixed and the carrier frequency component is filtered out to leave the baseband signal *B(t)* expressed by Equation (6), where $\lambda_{p}=\frac{c}{\sqrt{\in }}\cdot \frac{1}{f_{p}}$ is the wavelength of the emitted and scattered pulses in the subcutaneous tissue.
(6)$$B\left(t\right)=Bsin\left(\frac{4\pi \left(R_{o}-\Delta r\left(t\right)\right)}{\lambda_{p}}\right)$$

For hemodialysis patients, the distance $R_{o}$ between the AVF wall and skin is less than 6 mm and the diameter of the AVF is larger than 6 mm [16], with the periodical variation of the radius of the AVF, Δr estimated to be 0.028 mm [17]. Since $\Delta r\left(t\right)\ll R_{o}$, the motion of the AVF wall can be linearized to the following Equations (7) and (8), where $C_{1}$=$B(sin\left(\frac{4\pi }{\lambda_{p}}R_{o}\right)$ and $C_{2}=B\frac{4\pi }{\lambda_{p}}cos\left(\frac{4\pi }{\lambda_{p}}R_{o}\right)$ are constants.
(7)$$B\left(t\right)=B[\left(sin\left(\frac{4\pi }{\lambda_{p}}R_{o}\right)-\frac{4\pi }{\lambda_{p}}cos\left(\frac{4\pi }{\lambda_{p}}R_{o}\right)\left(\Delta r\left(t\right)\right)\right)\;$$
(8)$$\;\;\;\;\;\;\;B\left(t\right)=C_{1}-C_{2}*\left(\Delta r\left(t\right)\right)$$

Therefore, as derived in Equation (8), the baseband signal *B(t)* is linearly related to the displacement of the AVF wall $\Delta r\left(t\right)$.

The pulse radar sensor is fabricated on a flexible substrate (polyimide, size 8.0 × 3.5 cm, thickness 0.25 mm). Both antennas are planar microstrip monopoles. The input to each antenna is connected to a 50-ohm coplanar transmission microstrip line. The antenna input impedance matching was measured using a Vector Network Analyzer (Rohde & Schwartz, Muehldorfstrasse 15, 81671 Munich, Germany). The return loss was −20 dB at a resonant frequency of 1.38 GHz and the −10 dB bandwidth was 120 MHz. This lower antenna resonant frequency was chosen so that emitted pulses could penetrate the subcutaneous tissue surrounding the AVF vessel [18]. Since the antenna is working in the near field region, the medium between the sensor and skin surface is critical for coupling signal power through the skin barrier and subcutaneous tissues. Merli et al. [19] showed that the radiation efficiency of implanted antennas depended upon the dielectric properties of an insulating layer which separated the antenna from the surrounding muscle tissues. In the present work, it was found that a thin layer (0.25 to 1.0 mm) of textile material such as cotton or polyester cotton blended fiber is suitable for providing the desired properties. In addition to the insulating properties, the material’s biocompatibility and adherence to the skin are also important factors when selecting coupling materials.

The active low-pass filter smooths the mixed pulsatile signal into the baseband signal $B\left(t\right)$ (Figure 1A). The cutoff frequency of the active low pass filter is 25 Hz. The baseband signal is then amplified and fed into the MCU (Nordic Semiconductor, Trondheim, Norway), in which the A to D converter’s sampling frequency is set to 64 Hz. The digitized data are then transferred to a mobile phone using the built-in Low Energy Bluetooth Transceiver (LEBT) of the MCU. An App has been designed to display the signal waveform on the mobile phone during testing of the patient. To preserve the essential frequency contents of the baseband signal, a Hamming window-based bandpass, linear phase FIR filter has been designed using MATLAB 2020a (MathWorks, 1 Apple Hill Drive Natick, MA 01760, USA) in the frequency range of 0.2 to 10 Hz. The filtered data is then stored in the mobile phone and sent to a PC via a USB communication link for data analysis and classification.

Figure 2A shows baseband signals of three patients having normal AVF flows (1410, 1380, and 790 mL/min, respectively), in which stable and consistent VWM waveforms are evident. Figure 2B shows baseband signals of three patients having abnormal AVF flows (350, 430, and 360 mL/min, respectively), in which unstable and superimposed oscillations were observed on the VWM waveforms.

### 2.2. A Clinical Testing Protocol

The clinical trial was approved by the Institutional Review Board of Taichung Veterans General Hospital (TCVGH). A total of 46 patients regularly treated at the hemodialysis center in TCVGH were chosen for the clinical trial according to the inclusion criteria. There were 18 females with 67.9 ± 11.4 years old and 28 males with 61.9 ± 12.5 years old. One patient was excluded from the trial as his blood flow was not measurable due to difficulty in finding suitable sites for needle puncture. Informed consent was obtained from each patient prior to the start of the test session. The AVF locations on the tested patients were in various positions, from the wrist to the upper arm. The pulse radar sensor was attached according to each patient’s AVF location, and positioned near the venous outflow side distal to the AVF, as shown in Figure 3. The patient was instructed to remain still for 1 min with their arm supported on the table top. The acquired VWM data were wirelessly transferred using a Bluetooth transceiver in real time from the pulse radar sensor to the mobile phone, where the signal waveform was displayed and data stored. The stored data were then transferred from the mobile phone through the USB communication link to a laptop PC. The hemodialysis treatment was then applied to the same patient, with AVF flow measured by the UD flow instrument (HD03, Transonic Systems Inc., Ithaca, NY, USA) within the first 30 min of the hemodialysis treatment session. The measured flow data was recorded and stored in a laptop PC for later analysis.

### 2.3. Data Processing and SVM Classification

The VWM data were converted to spectrum data using the Fast Fourier Transform algorithm in MATLAB 2020a. Due to its advantage in classifying small-sized complex datasets, the SVM machine learning classification algorithm was developed to predict the patient’s AVF status using the features derived from the FFT spectrum data. The Radial Basis Function (RBF) kernel was chosen to train the SVM algorithm for classification of the non-linear datasets. The cutoff value between the abnormal and normal data sets was chosen according to the suggestions of the NKF-K/DOQI guidelines with a threshold of 600 mL/min and followed up in the hemodialysis center in TVGH for the detection of flow dysfunction. In addition to this regularly used cutoff value, the ROC analysis was used to verify the performance of the SVM classifier and to search for an optimal threshold in the detection of flow dysfunction.

The spectral (frequency-domain) analysis of physiological variables, using the fast Fourier transformation (FFT), has been reported [20]. A total of five features was used in this study, which are the ratios of FFT spectral peaks of the higher harmonics to those of the nearest lower harmonics, defined as the Harmonic Ratio (HR), i.e., P2/P1, P3/P2, P4/P3, P5/P4, and P6/P5. For example, Figure 4 compares the FFT spectrum of a normal VWM signal (left panel) with that of an abnormal VWM signal (right panel), in which a distinct difference exists in the ratio of P3 to P2. To validate the HRs as being features for VSM classification, the mean differences between the HRs of abnormal FFT spectrums and those of normal FFT spectrums were tested using the independent T-test (α = 0.05). Table 1 shows that P5/P4 is a sensitive feature for a detection threshold of 600 mL/min (*p* = 0.008), and that P3/P2 is a sensitive feature for a detection threshold of 750 mL/min (*p* = 0.041). Based on this sensitivity analysis, the HRs for each patient were chosen to be the input features for the SVM classification algorithm.

## 3. Results

Figure 5 displays the results of training the SVM classifier with detection thresholds of 600 mL/min. Each data point in the figure represents a set of two values in which the correponding value on the *x* axis is the value of the decision function, while that on the *y* axis is the measured flow data by UD. The value of the decision function shows whether an output by the SVM classifier lies to the right or left side of the hyperplane (*y* axis), as well as how far it is from the hyperplane. The hyperplane is an optimal plane separating the two classes with maximum margin. When the output value of the decision function is close to zero on the hyperplane, it represents a low-confidence decision, whereas when the output values of the decision function are a larger magnitude of positive or negative values, the more confident the decisions are. The locations of the data points relative to both the detection threshold (horizontal red line) and hyperplane determine whether the classification results are true or false. As defined by the hyperplane and detection threshold, when a data point is located either in the lower left or upper right region, it is being classified correctly as true positive or true negative, respectively. Alternatively, when a data point is located either in the upper left or lower right region, it is being classified incorrectly as fase positive or false negative, respectively. For a detection threshold of 600 mL/min, the outputs of the SVM classifier were mostly located in the true positive and true negative regions, with no false positive predictions and only one false negative being classified (shown in Figure 5).

The trained SVM classifier was validated using the method of 10-fold cross validation. Table 2 summarizes the validated results of the VSM classifier with a detection threshold of 600 and 750 mL/min. The performance of the SVM classifier shows a sensitivity of 90.9% (95.2%), specificity of 100.0% (100.0%), and an accuracy of 97.8% (97.8%) for 600 (750) mL/min. Notice that the prediction accuracy is 100.0% if measured by the positive prediction value.

As mentioned in the NKF-K/DOQI guidelines, there exists a need for standardized diagnostic thresholds with sufficient sensitivity as well as specificity. In the present work, the ROC curve analysis was performed with the results showing that the Area Under the Curve (AUC) was 0.994. The specificity and positive prediction value for the detection thresholds of 750, 650, 600, and 500 mL/min were all 100%, with the maximum sensitivity being 95.2% at a detection threshold of 750 mL/min (shown in Figure 6). This result indicates that the VWM monitor-based AVF flow dysfunction detector provides an excellent correlation with the UD flow monitor within a large range of AVF flow. However, a single optimal detection threshold could not be determined solely on the value of sensitivity. Instead, two levels of detection thresholds, e.g., firth level at 750 mL/min and second level at 600 mL/min, may be more reliable for the early detection of flow dysfunction before proceeding to pre-emptive angioplasty.

## 4. Discussion

The wall motion in a stenotic carotid artery was investigated by Kanber et al. [21]. Due to the difficulty to measure stable signals in the stenotic region, the proximal shoulder of the atherosclerotic region was chosen as the measurement site, with ultrasound image sequences being acquired over several cardiac cycles. Results showed that both absolute and percentage diameter changes did not have any statistically significant relationship to the degree of stenosis. In the present work, measurement sites on patients’ arms were all located at the venous outflow side of AVFs distal to where the periodical generation of flow turbulence took place. The signals acquired by the VWM monitor showed differences in waveform morphology between the abnormal flow cases and those of the normal flow cases. Figure 7A shows an example of a distorted AVF VWM signal (low flow, flow = 360 mL/min) whose waveform morphology displayed oscillations superimposed on one cycle of the original AVF VWM waveform. This is believed to be the result of the modulation of two signals, i.e., the original VWM signal and the signal with oscillations due to flow turbulence. Consequently, as shown in Figure 7B, new harmonics P3 and P5 appear with higher strengths in the FFT spectrum than those of the original VWM signal. Since the oscillating frequency is three to five times the fundamental frequency, the modulation effect would increase the strength of the higher frequency harmonics. This observation could be the basis for future work on detection of flow dysfunction by combining time domain and frequency domain analytical techniques.

Tessitore et al. [5] investigated the optimal thresholds for stenosis detection using the UD flow measurement. Their results showed that a detection threshold of 750 mL/min was optimal for AVFs at the wrist and 1000 mL/min for AVFs at the mid-forearm. In the present work, 50% of AVFs were located on patients’ mid-forearms, 40% on patients’ wrists, and the remaining 10% on patients’ elbows. The ROC analysis results of the SVM classifier found that maximum sensitivity was achieved at a detection threshold of 750 mL/min, which was identical to that reported in [5] on stenosis detection. In the future, through the use of fistulography as the gold standard, an expanded clinical study will be needed in order to validate the SVM classifier in predicting stenosis.

## 5. Conclusions

In conclusion, this study is based on previous studies which demonstrated that stenosis of the arterial blood vessel could be detected by analyzing the characteristic changes in the waveform of VWM. In this work, the operating principle of the pulse radar sensor for detection of VWM was derived theoretically and the performance of the VWM monitoring system was verified clinically. The VWM monitoring system was applied to detect the flow dysfunction in AVFs on patients who were receiving the hemodialysis treatment. Harmonic ratios derived from the FFT spectrum of the VWM monitoring signals were used as the input features to a SVM classification algorithm. Ten-fold cross validation results revealed an excellent correlation between the VWM monitor and UD flow monitor. To ensure the operation reliability of the VWM monitoring system, the long-term reproducibility of the as developed VWM monitoring system will be evaluated in the near future.

By adapting the two-level detection threshold method for early detection, the VWM monitoring technique for self-tests at home, or regular screening in hemodialysis centers, can provide the benefits of both reducing the present workload in testing AVF flow dysfunction in the hospital, as well as assuring the quality of care needed to preserve AVF patency. Meanwhile, the long-term reproducibility of the as developed VWM monitoring system should be evaluated in the near future.

## Figures and Tables

**Figure 1 biosensors-11-00297-f001:**
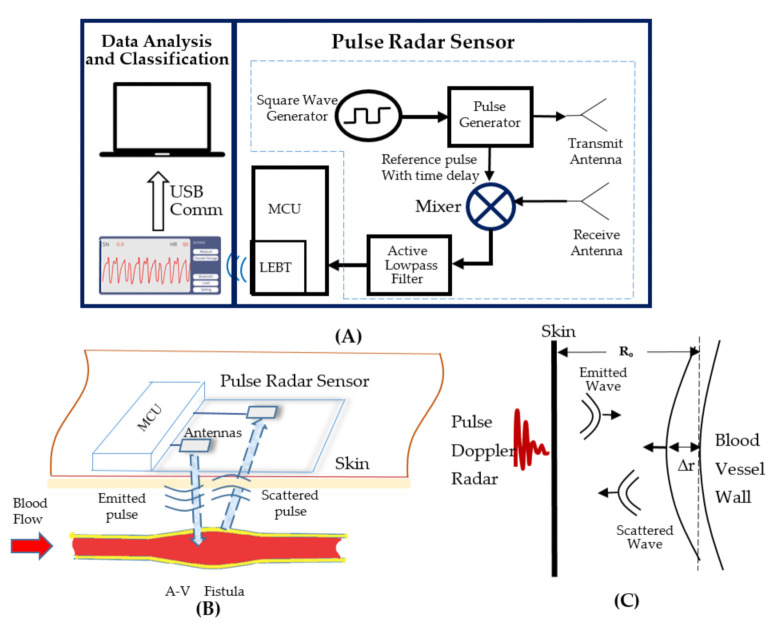
The system diagram of the VWM monitor and the principle of measuring the AVF vessel wall movement (**A**). The diagram of pulse radar sensor detect blood flow (**B**). The diagram of pulse Doppler radar emit and receive the signals of wave (**C**).

**Figure 2 biosensors-11-00297-f002:**
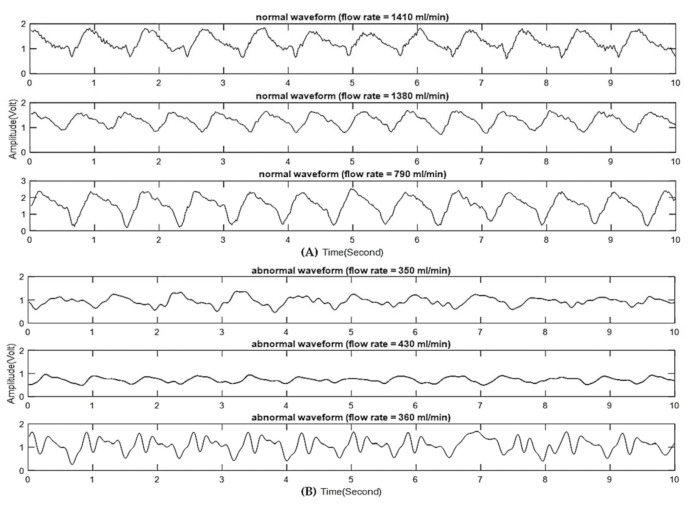
(**A**) Baseband signals of normal AVF with stable and consistent VWM waveforms. (**B**) Baseband signals of abnormal AVF with unstable and superimposed oscillations on VWM waveforms.

**Figure 3 biosensors-11-00297-f003:**
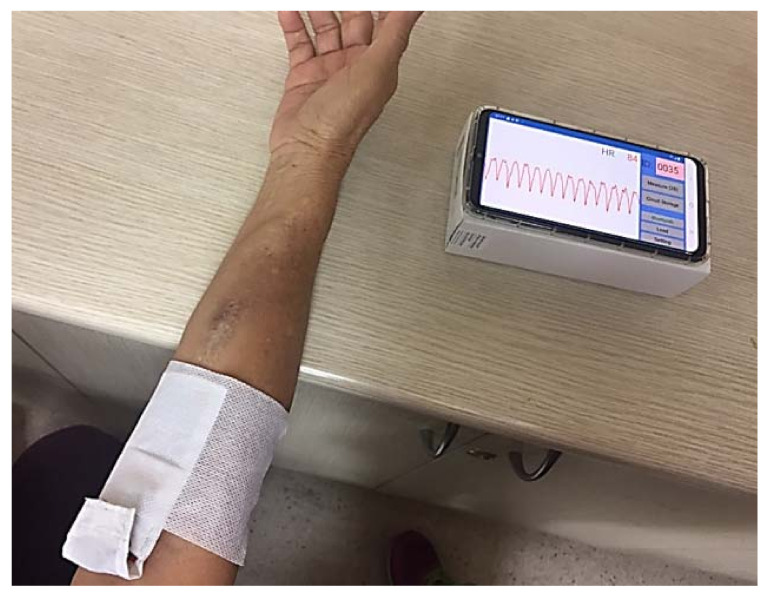
The hemodialysis patient during testing using a pulse radar sensor. Note that the App displays the signal waveform on a mobile phone. The detailed strategy of immobilizing as developed by the VWM monitoring system on patients, shown in Appendix A.

**Figure 4 biosensors-11-00297-f004:**
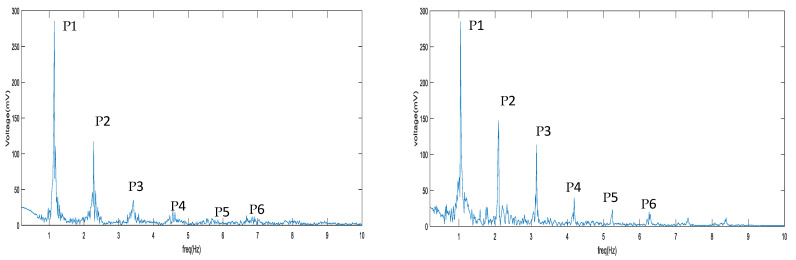
FFT spectrum of normal AVF wall motion signal (left), FFT spectrum of abnormal AVF wall motion signal (right).

**Figure 5 biosensors-11-00297-f005:**
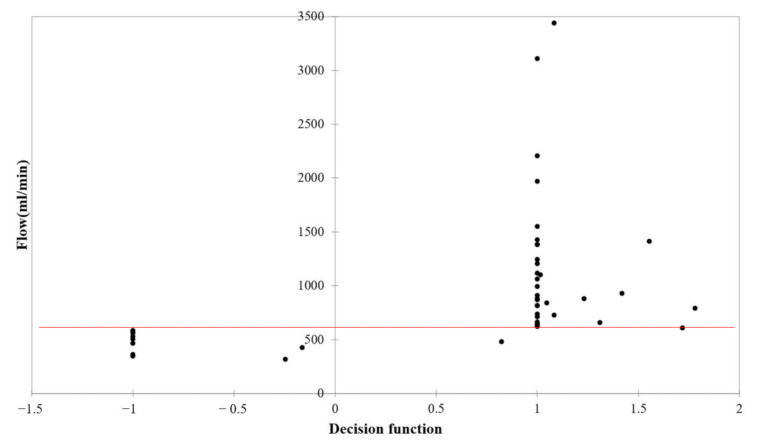
Results of SVM classifier training with a detection threshold of 600 mL/min.

**Figure 6 biosensors-11-00297-f006:**
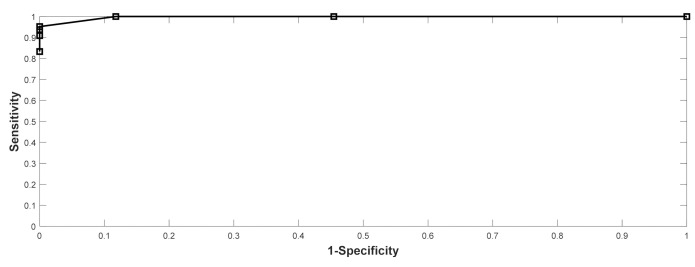
The results of ROC analysis on the performance of the SVM classifier.

**Figure 7 biosensors-11-00297-f007:**
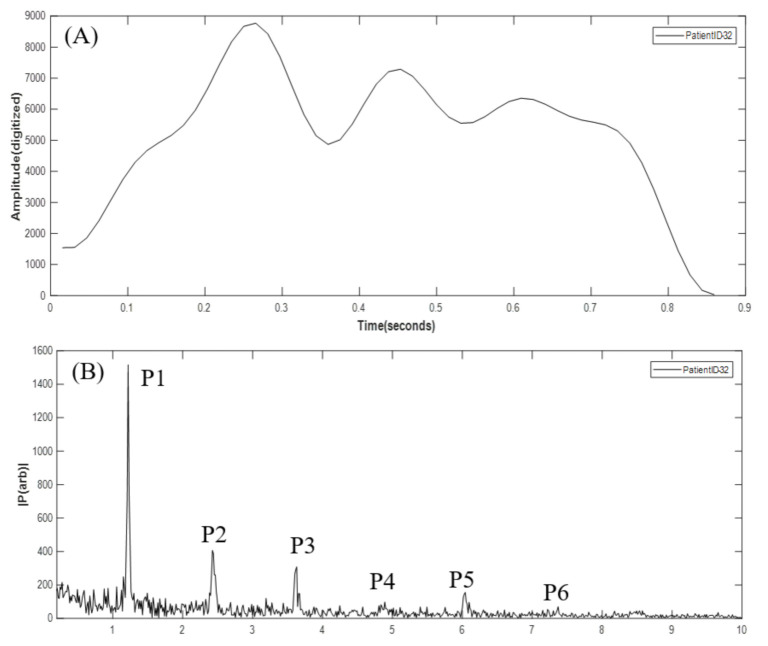
(**A**) Abnormal oscillations superimposed on one cycle of the VWM waveform. (**B**) The abnormal FFT spectrum with more pronounced spectral peaks at P3 and P5.

**Table 1 biosensors-11-00297-t001:** Independent T-test of the mean difference in harmonic ratios between low flow cases and high flow cases relative to cutoff values at 600 and 750 mL/min (referenced in Figure 4). Notice symbol * represents the *p*-value < 0.05.

Harmonic Ratio	≤600 (n = 11)	>600 (n = 34)	Difference	*p*-Value	≤750 (n = 21)	>750 (n = 24)	Difference	*p*-Value
	Mean (SD)	Mean (SD)			Mean (SD)	Mean (SD)		
P2/P1	0.357 (0.203)	0.380 (0.178)	−0.023	0.722	0.347 (0.168)	0.398 (0.195)	−0.051	0.356
P3/P2	0.665 (0.313)	0.491 (0.285)	0.174	0.093	0.630 (0.277)	0.449 (0.296)	0.181	0.041 *
P4/P3	0.591 (0.289)	0.785 (0.456)	−0.193	0.195	0.774 (0.475)	0.705 (0.387)	0.069	0.591
P5/P4	1.075 (0.601)	0.711 (0.273)	0.365	0.008 *	0.818 (0.510)	0.784 (0.293)	0.035	0.777
P6/P5	0.659 (0.235)	0.751 (0.347)	−0.092	0.417	0.696 (0.352)	0.758 (0.300)	−0.062	0.529

**Table 2 biosensors-11-00297-t002:** The 10-fold cross validation results for SVM classifier at a detection threshold of 600 and 750 mL/min, respectively.

		Ground Truth				
Threshold	Prediction	Flow ≤ 600	Flow > 600	Total	%Correct	Index
**600**	**Flow ≤ 600**	10	0	10	90.9	Sensitivity
	**Flow > 600**	1	34	35	100.0	Specificity
	**Total**	11	34	45	97.8	Accuracy
**Threshold**	**Prediction**	**Flow ≤ 750**	**Flow > 750**	**Total**	**%Correct**	**Index**
**750**	**Flow ≤ 750**	20	1	21	95.2	Sensitivity
	**Flow > 750**	0	24	24	100.0	Specificity
	**Total**	20	25	45	97.8	Accuracy

## Data Availability

From 11 September 2019 to 10 September 2020.

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
