# Peer review of "Arteriovenous Fistula Flow Dysfunction Surveillance: Early Detection Using Pulse Radar Sensor and Machine Learning Classification"

_biosensors, 2021, doi:10.3390/bios11090297_

Round 1

Reviewer 1 Report

In this manuscript, the authors developed a non-invasive Vascular Wall Motion (VWM) monitoring system for detecting access flow dysfunction in Arteriovenous Fistula (AVF) by made up of a pulse radar sensor and Support Vector Machine (SVM) classifica-tion algorithm. The as-developed VWM monitoring system exhibits a detection specificity of 100% at detection thresholds in the range from 500ml/min to 750ml/min and a maximum sensitivity of 95.2% at a detection threshold of 750ml/min. Specific comments,

(1) The cost of pulse Radar sensor should be estimated.

(2) The long-term reproducibility of as-developed VWM monitoring system should be evaluated.

(3) The personal details (e.g., age, bodyweight, male/female, etc.) of 46 patients should be included in the manuscript or indicated as supporting information.

(4) The English writing should be checked carefully, such as ‘Onepatient’, ‘Figure.3’, etc.

(5) The detailed strategy of immobilizing as-developed VWM monitoring system on patients should be shown.

Author Response

Response to Reviewer 1 Comments

In this manuscript, the authors developed a non-invasive Vascular Wall Motion (VWM) monitoring system for detecting access flow dysfunction in Arteriovenous Fistula (AVF) by made up of a pulse radar sensor and Support Vector Machine (SVM) classification algorithm. The as-developed VWM monitoring system exhibits a detection specificity of 100% at detection thresholds in the range from 500 ml/min to 750 ml/min and a maximum sensitivity of 95.2% at a detection threshold of 750 ml/min. Specific comments,

Point 1: The cost of pulse Radar sensor should be estimated.

Response 1: Thanks reviewer suggestion and comments.

The cost of pulse radar sensor is around $150 USD.

Point 2: The long-term reproducibility of as developed VWM monitoring system should be evaluated.

Response 2: Thanks reviewer suggestion and comments.

The long-term reproducibility of as developed VWM monitoring system should be evaluated in the near future. The sentence has been added in this revision.  

Point 3: The personal details (e.g., age, bodyweight, male/female, etc.) of 46 patients should be included in the manuscript or indicated as supporting information.

Response 3: Thanks reviewer suggestion and comments.

There were 18 females with 67.9 ± 11.4 years old and 28 males with 61.9 ± 12.5 years old in this study. This sentence had been added in Section 2.2.

Point 4: The English writing should be checked carefully, such as ‘Onepatient’, ‘Figure.3’, etc.

Response 4: Thanks reviewer suggestion and comments.

The English writing and typos are fixed already in this revision.

Point 5: The detailed strategy of immobilizing as developed VWM monitoring system on patients should be shown.

Response 5: Thanks reviewer suggestion and comments.

The detailed strategy of immobilizing is shown in Appendix A for the VWM monitoring system on patients.

Reviewer 2 Report

It is an interesting topic and could be useful in real life situations. However, lot of improvements are needed. Below points need to be considered

  • It was not clear how many features are extracted and given as input to the SVM classifier?
  • Are all the features those are extracted are useful for classification? Any reference papers related to this work should be cited in this paper
  • If possible different feature selection technique should be tried to find the ranking of the features.
  • Why the author chooses only the SVM classifier for this work. Is this the only classifier which gives better result? Detail explanation is required. If possible, add other classifiers and do the comparison of the results.
  • Cross validation technique needs to be used to validate the result? K-fold cross validation could be a better choice.
  • Confusion matrices need to be added in this paper.
  • The conclusion part needs better development. Research implications theoretical & practical should be clearly presented with reminding us of the key contribution of the research.

Author Response

Response to Reviewer 2 Comments

It is an interesting topic and could be useful in real life situations. However, lot of improvements are needed. Below points need to be considered

Point 1: It was not clear how many features are extracted and given as input to the SVM classifier?

Response 1: Thanks reviewer suggestion and comments.

Total of five features were used in this study which were the ratios of FFT spectral peaks of the higher harmonics to those of the nearest lower harmonics, defined as the Harmonic Ratio (HR), i.e. P2/P1, P3/P2, P4/P3, P5/P4, P6/P5 added in Section 2.3 in this revision.   

Point 2: Are all the features those are extracted are useful for classification? Any reference papers related to this work should be cited in this paper. If possible different feature selection technique should be tried to find the ranking of the features.

Response 2: Thanks reviewer suggestion and comments.

Total of five features were used in this study for SVM. The related reference was cited as [20] in this revision.

[20] Pybus, DA. Real-time, spectral analysis of the arterial pressure waveform using a wirelessly-connected, tablet computer: a pilot study. J Clin Monit Comput. 2019, 33(1), 53-63.

Also, the feature selection technique was applied Information Gain Attribute Evaluation Method in Weka 3.8.4. The ranks of used attribute (feature) were shown in the following Table. The five features are adopted for SVM and generated useful classified results. Therefore, the five features are utilized in this study.

Attribute

Average Score

Rank

A1

1.4±1.2

1

A2

2.0±0.0

2

A5

2.8±0.6

5

A4

4.0±0.0

4

A3

4.8±0.6

3

Point 3: Why the author chooses only the SVM classifier for this work. Is this the only classifier which gives better result? Detail explanation is required. If possible, add other classifiers and do the comparison of the results.

Response 3: Thanks reviewer suggestion and comments.

The SVM is provided better validated consequence than the other classifiers in this work. Meanwhile, the classification by SVM is suitable for the developed VWM monitoring system. Hence, the results provided by SVM was demonstration in this study.

Point 4: Cross validation technique needs to be used to validate the result? K-fold cross validation could be a better choice.

Response 4: Thanks reviewer suggestion and comments.

The K-fold cross validation had been applied in this study already. In this study, the K is 10.

In Table 2, the 10-fold cross validation results for SVM classifier at a detection threshold of 600 and 750 ml/min, respectively.

Point 5: Confusion matrices need to be added in this paper.

Response 5: Thanks reviewer suggestion and comments.

The 10-fold cross validation with confusion matrices had been applied in Table 2 in this revision.

Point 6: The conclusion part needs better development. Research implications theoretical & practical should be clearly presented with reminding us of the key contribution of the research.

Response 6: Thanks reviewer suggestion and comments.

The conclusion part had been modified and added the key contribution of the research in this revision as below.

Based on previous studies which demonstrated that stenosis of the arterial blood vessel could be detected by analysing the characteristic changes in the waveform of VWM. In this work, the operating principle of the pulse radar sensor for detection of VWM was derived theoretically and the performance of the VWM monitoring system was verified clinically. The VWM monitoring system was applied to detect flow dys-function in AVFs on patients who were receiving hemodialysis treatment. Harmonic ratios derived from the FFT spectrum of the VWM monitoring signals were used as the input features to a SVM classification algorithm. The performance of the VWM monitoring system was verified through a pilot clinical trial. The harmonic ratios derived from the FFT spectrum of the VWM monitoring signals were used as the input features to a SVM classification algorithm. Ten-fold cross validation results revealed an excel-lent correlation between the VWM monitor and UD flow monitor. To ensure the opera-tion reliability of the VWM monitoring system, long-term reproducibility of as developed VWM monitoring system will be evaluated in the near future.

By adapting the two-level detection threshold method for early detection, the VWM monitoring technique for self-tests at home, or regular screening in hemodialysis centres, can provide the benefits of both reducing the present workload in testing AVF flow dysfunction in the hospital, as well as assuring the quality of care needed to pre-serve AVF patency. Meanwhile, the long-term reproducibility of as developed VWM monitoring system should be evaluated in the near future”.  
